# Hydroxyapatite/Poly (Butylene Succinate)/Metoprolol Tartrate Composites with Controllable Drug Release and a Porous Structure for Bone Scaffold Application

**DOI:** 10.3390/polym15214205

**Published:** 2023-10-24

**Authors:** Hongming Yang, Rui Pan, Yuan Zhou, Guiting Liu, Rong Chen, Shaoyun Guo

**Affiliations:** The State Key Laboratory of Polymer Materials Engineering, Polymer Research Institute of Sichuan University, Chengdu 610065, China; 2022223095220@stu.scu.edu.cn (H.Y.); 14726015010@163.com (R.P.); zhouyuan1@stu.scu.edu.cn (Y.Z.); nic7702@scu.edu.cn (S.G.)

**Keywords:** composites, hot processing, scaffold

## Abstract

Nowadays, it is a challenge for a bone scaffold to achieve controllable drug release and a porous structure at the same time. Herein, we fabricated hydroxyapatite/poly (butylene succinate)/metoprolol tartrate (HA/PBS/MPT) composites via melt blending, aiming to provide the option of an in situ pore-forming strategy. The introduction of HA not only significantly improved the hydrophilicity of the PBS matrix by reducing the hydrophilic contact angle by approximately 36% at a 10% content, but also damaged the integrity of the PBS crystal. Both were beneficial for the penetration of phosphate-buffered saline solution into matrix and the acceleration of MPT release. Accompanied with MPT release, porous structures were formed in situ, and the HA inside the matrix was exposed. With the increase in HA content, the MPT release rate accelerated and the pore size became larger. The in vitro cytocompatibility evaluation indicated that HA/PBS/MPT composites were conductive to the adhesion, growth, and proliferation of MC3T3-E1 cells due to the HA being exposed around the pores. Thus, the MPT release rate, pore size, and cell induction ability of the HA/PBS/MPT composites were flexibly and effectively adjusted by the composition at the same time. By introducing HA, we innovatively achieved the construction of porous structures during the drug release process, without the addition of pore-forming agents. This approach allows the drug delivery system to combine controllable drug release and biocompatibility effectively, offering a novel method for bone repair material preparation. This work might provide a convenient and robust strategy for the fabrication of bone scaffolds with controllable drug release and porous structures.

## 1. Introduction

Bone repair has always been one of the most significant global health challenges in the past few decades [1]. Bone transplantation is the commonly utilized and efficient clinical treatment for bone repair. However, there are still some severe problems in transplantation treatment, such as a limited source of autologous bone, immune rejection, the potential virus transmission of allogeneic bone, and postoperative complications [2]. Recently, bone tissue engineering scaffolds have provided a new and promising alternative strategy for bone repair, as their porous structures could simulate extracellular matrix and promote the migration, proliferation, and differentiation of osteoblasts, so as to accelerate the repair of bone defects [3,4]. In addition, bone scaffolds are usually designed to degrade completely and have very little impact on bodily health [5]. With the advancement of their clinical application, bone scaffolds are often required to play the role of carriers to deliver bioactive agents to enhance curative effect, such as antihypertensive, hypoglycemic, and anti-inflammatory drugs, among others [6,7]. Thus, this puts forward new requirements for bone scaffolds to combine material selection, porous structures, and drug release.

Hydroxyapatite (HA) is an important mineral component in human and animal bones, and it can promote the proliferation and differentiation of osteoblasts [8]. At present, HA has developed into one of most frequently used materials for bone scaffolds because of its excellent bioactivity, biocompatibility, and bone conduction [9]. In most cases, HA is compounded with polymer materials, especially biodegradable polymers (BP), to overcome the disadvantages of its poor toughness and difficult molding [10]. Additionally, compared to HA scaffolds, HA/BP scaffolds are closer to natural bone in terms of structure and composition, and have better osteogenic properties [11]. For traditional non-porous HA/BP composite scaffolds, HA is imbedded in the polymer matrix and has a low utilization and lack of bone conduction. Thus, various pore-forming methods, such as foam impregnation, the addition of porogen, and the sol–gel method, have been developed to regulate the porous structures (pore size, porosity, and pore connectivity) of HA/BP composite scaffolds and make a significant achievement in bone regeneration [12,13]. However, endowing HA/BP porous scaffolds with controllable drug release behavior remains a tough challenge. The pore forming via the current method is usually carried out after the loading of the drug, damaging the uniform distribution of the drug in the scaffold and leading to unreliable drug release behavior. Therefore, it is urgent for HA/BP scaffolds to realize controllable porous structures and drug release.

Herein, for the first time, we attempt to develop an in situ pore-forming method to fabricate HA/BP scaffolds with a controllable porous structure and drug release. The “in situ pore formation technique” allows for the creation of pores or void spaces within a material during its formation or fabrication process, rather than introducing them through post-processing methods afterward. Poly (butylene succinate) (PBS), a biodegradable and biocompatible polymer, is selected as the scaffold material [14], with metoprolol tartrate (MPT), widely used for delivering bioactive molecules that promote bone cell proliferation and bone tissue regeneration, thereby aiding in the acceleration of bone fracture healing and bone defect repair. Meanwhile, the mechanism of action of MPT has become increasingly mature. Thus, MPT is often used as a drug model for studying various types of drug release systems. By studying the release behavior of MPT, the performance and efficiency of drug delivery systems can be assessed. HA/PBS/MPT composites are constructed by melt blending. In this design, the introduction of HA could not only significantly improve the hydrophilicity of the PBS matrix, but also damage the integrity of the PBS crystal. Both were beneficial to the penetration of phosphate-buffered saline solution into the matrix and accelerated the MPT release. The porous structure of the matrix could be formed in situ after the release of the MPT. Thus, the porous structure and MPT release of HA/PBS/MPT scaffolds could be flexibly and effectively regulated by the composition at the same time (Figure 1). The use of PBS combined with HA in the preparation of bone repair or tissue-engineering scaffold materials enables the construction of pore structures during the drug release process. This eliminates the need for separate local drug administration at the injury site and can achieve pore structure construction without the addition of other pore-forming agents. It fully leverages the biocompatibility and bone-promoting effects of HA, making the HA/PBS/MPT composite material a promising candidate for achieving synergistic therapy involving drug release and bone formation induction when used as a bone scaffold.

## 2. Materials and Methods

### 2.1. Materials

Poly (butylene succinate) (PBS), with melt index (MI) of 18 g/10 min (190 °C/2.16 kg), was purchased from Anqing Hexing Chemical Co., Ltd. (Anqing, China). Metoprolol tartrate (MPT) was obtained from Guangzhou Baiyunshan Hanfang Modern Pharmaceutical Co., Ltd. (Guangzhou, China).

Hydroxyapatite (HA), with particle size of 20 nm and a purity of 99%, was supplied by Nanjing April nano materials Co., Ltd. (Nanjing, China). Phosphate-buffered saline (PBS, pH 7.4) was obtained from Zhongshan Golden Bridge Biotechnology Corp., (Beijing, China). Both Rhodamine tetramethylisothiocyanate labeled ghost pen cyclic peptide (TRITC phalloidin) and 4′, 6-diamidino-2-phenylindole dihydrochloride (DAPI) were supplied by Beijing Solarbio Technology Co., Ltd. (Beijing, China).

### 2.2. The Preparation of Composite Samples

The PBS, MPT, and HA were dried in a vacuum oven at 40 °C. After 24 h, they were taken out and weighed in proportion. An Rm-200c torque rheometer was used for the internal mixing. The internal mixing temperature was set at 130 °C, the rotating speed was 50 rpm, and the internal mixing time was 8 min. After the internal mixing, the samples were dried in a vacuum oven (40 °C) for 24 h. To ensure the uniformity of the sample thickness, the samples were then hot pressed at 130 °C for 5 min with an R-3212 hot press (Wuhan Qien Technology Development Co., Ltd., Wuhan, China) at a pressure of 10 MPa. Finally, the samples were cold pressed to room temperature under the same pressure to form a disc with a diameter of 20 mm and a thickness of 1 mm. The descriptors and composition of the composite samples are presented in Table 1.

### 2.3. The Characterization of Composite Samples

A thermogravimetric analysis (TGA) was performed with a TA 209 F1 thermal analyzer (Netzsch Company, Bavaria, Germany) in a nitrogen atmosphere. The temperature was raised from 30 °C to 600 °C with a heating rate of 10 °C/min, and the mass loss of the samples during the heating process was recorded. FTIR spectra of the samples were obtained with a Fourier transform infrared spectrometer (IS10, Thermo Nicolet Company, Madison, WI, USA) in the region of 400–4000 cm^−1^. Polarizing microscope (POM) photos were obtained with a Bx-51 polarizing microscope (Olympus Company, Tokyo, Japan), samples were taken along the thickness direction of the samples, and the slice thickness was set to 15 μm.

For a differential scanning calorimetry (DSC) analysis, the samples were tested using a Q20 differential scanning calorimeter. For each test, the temperature was set from −60 °C to 150 °C at a temperature rise rate of 10 °C/min in a nitrogen (flow rate of 50 mL/min) atmosphere, isothermal for 3 min to eliminate the thermal history, then the temperature was reduced to −60 °C at a temperature drop rate of 10 °C/min to prevent an inconsistent internal crystalline orientation of the PBS and MPT during the blending processing. After that, the first temperature rise process was repeated. Both the curve of the cooling process and the curve of the second heating process were recorded.

A DSA25S contact angle measuring instrument (Kruss Company, Hamburg, Germany) was used to test the hydrophilic contact angle on the sample surfaces. Each sample was tested 6 times, then the average value was taken. The morphology of the samples was observed using a KL30FZG scanning electron microscope (Philips Company, Amsterdam, The Netherlands). Before each test, the samples were quenched in liquid nitrogen to obtain brittle fracture surfaces for better cross-sectional morphology, and then the fracture surfaces were subjected to a vacuum gold coating treatment. The pore size of the samples was analyzed and counted using the Image J software (ImageJ 1.53, National Institutes of Health, Bethesda, MD, USA).

### 2.4. In Vitro Drug Release Test

The cumulative drug release test was measured using a UV-1750 ultraviolet visible spectrophotometer (UV-1750, Shimadzu company, Kyoto, Japan). The characteristic absorption peak of the MPT in the UV spectrum was 222 nm, and the applicable concentration range of the standard curve for the determination of the MPT was 0.6–60 μg/mL. There was a good linear relationship between the UV absorbance and MPT concentration in this range (r > 0.999).

For each test, the samples were immersed in 10 mL of phosphate-buffered saline (PBS) solutions and incubated in an incubator (100 rpm, 37 °C). At each time point selected for measurement, the solution was totally exchanged with fresh PBS. All the experiments were performed in triplicate. The concentrations of MPT were calculated according to the standard curve equation.

### 2.5. In Vitro Cytocompatibility Evaluation

Before evaluation, the samples were immersed in PBS solution for 5 days. Then, they were taken out and washed with deionized water at least three times, followed by dry treatment in a vacuum oven (40 °C) for further application. For a cell viability evaluation, the samples were co-cultured with MC3T3-E1 cells in 96-well plates (6 × 10^3^ cells/well) and placed in a cell incubator (Thermo Fisher, Wilmington, MA, USA) with a temperature of 37 °C and 5% CO_2_ content. After 24 h of culturation, cytoskeletal staining (TRITC Phalloidin/DAPI staining) was also utilized to investigate the cytocompatibility of the samples. The staining was processed according to the previous literature [15,16], and images of the cells were observed with an inverted fluorescence microscope (Leica DMi8 high-speed imaging platform, Wetzlar, Germany).

### 2.6. Statistics Analysis

GraphPad Prism 8 software was used for the statistical analysis. The obtained data were analyzed using a *t*-test. All the results were expressed in the form of mean ± standard deviation. Values were considered statistically significant with *p* < 0.05 and *p** represented *p* < 0.05.

## 3. Results

### 3.1. Thermal and Component Stability during Heat Treatment

The thermogravimetric (TGA) curves of the samples are shown in Figure 2a, and the initial decomposition temperature and residue mass ratio of the samples are listed in Table 2. It can be observed that the initial decomposition temperature of the PBS was approximately 348.4 °C, and the residue was 0.3 wt%, indicating a complete degradation of the PBS. For the PBS/HA samples, the initial decomposition temperatures of HA-2%, HA-5%, and HA-10% were 355.1 °C, 358.1 °C, and 359.9 °C, respectively. With an increase in the HA content, the initial decomposition temperature of the PBS/HA samples increased, and the residue was equivalent to the weight of the HA, indicating that the HA underwent minimal degradation at 600 °C. The degradation of the MPT-HA-10% sample could be divided into three stages. The first stage was the decomposition of the MPT, which started at about 181 °C, and the mass loss rate was about 10%, which was equivalent to the content of the MPT. The second stage was the degradation of the PBS, which started at about 350 °C and ended at about 410 °C. The mass loss rate was about 80%, which was equivalent to the content of the PBS. The third stage was the residue of the HA, which nearly did not degrade. It was shown that the initial decomposition temperature of each sample was higher than that used in the internal mixing and hot pressing processes (130 °C), indicating the thermal stability of the components during the internal mixing and hot pressing processes.

The FTIR spectra of the samples are presented in Figure 2b,c. The absorption peak of HA at 3446 cm^−1^ was assigned to the stretching peak of the O-H bond, and the absorption peaks at 1045 cm^−1^, 960 cm^−1^, 609 cm^−1^, and 570 cm^−1^ were attributed to the characteristic absorption peaks of PO_4_^3−^. The absorption peak of PBS at 1735~1711 cm^−1^ belonged to the stretching vibration of the C=O bond, the absorption peak near 1050 cm^−1^ was assigned to the characteristic absorption peak of the C-O bond in the PBS hydroxyl group, and the absorption peak near 1280 cm^−1^ was attributed to the characteristic absorption peak of the C-O bond in the PBS carboxyl group. PBS was the main component in the PBS/HA and HA/PBS/MPT composites. The infrared spectrum mainly showed the absorption peak of PBS, indicating that there was no obvious interaction and covalent bond between the PBS and HA [17]. In addition, the UV characteristic absorption peaks of MPT before and after loading in the scaffold were located at 222 nm (Figure 2d), showing no obvious peak shape change and peak position shift, indicating that the chemical structure of MPT did not change significantly after the heat treatment.

### 3.2. The Morphology and Hydrophilic Contact Angle Test

The section morphology of the PBS and PBS/HA composites with different HA contents is shown in Figure 3. It can be seen that the morphology of PBS was smooth and dense, while that of PBS/HA was dense and there was obvious HA particle aggregation. With an increasing in the HA content, the aggregation of HA became more serious. The aggregation of HA within a polymer matrix is a common phenomenon, which can result in greater restrictions on the diffusion and release of drug molecules, leading to a decrease in the drug release rate. However, at lower HA concentrations, the hydrophilicity of HA has a more pronounced effect on the enhancement of the drug release rate. The hydrophilic contact angle test was used to characterize the hydrophilicity of the MPT-10% and HA/PBS/MPT composites. As indicated in Figure 3i, the hydrophilic contact angle of MPT-10% was 61°, and the hydrophilicity of the HA/PBS/MP composite gradually increased with an increase in the HA content. For MPT-HA-10%, the water contact angle was 24.9°, which was about 36.1° lower than that of MPT-10%.

### 3.3. Crystal Morphology of Samples Evaluation

The polarizing microscope (POM) photos of the samples are shown in Figure 4a–g. Clear and uniform ring spherulite was captured in the PBS samples. With an increase in the HA content, the crystalline morphology of the PBS was affected due to the steric effect of HA. The steric effect of HA refers to the phenomenon where the introduction of HA molecules or particles disrupts the normal arrangement and crystallization process of the polymer chains or lattice. In the PBS/MPT/HA system, the crystal integrity of PBS was even more compromised due to the increased steric hindrance caused by MPT and HA compared to the PBS/HA system. This will lead to an improvement in the permeability of the matrix, thereby promoting drug release. The DSC curves of the samples in the cooling process after eliminating the thermal history and in the heating process are presented in Figure 4h,i, respectively. The crystallization temperature, melting point, melting enthalpy, and crystallinity of the samples were calculated and are listed in Table 3, respectively. Due to the addition of HA, the crystallization of PBS exhibited more defects, facilitating the transition from solid to liquid, which resulted in a decrease in the crystallization temperature, melting point, ΔHm, and crystallinity of the samples. It could be concluded that the introduction of HA decreased the crystal integrity of PBS, which was consistent with the POM observation.

### 3.4. In Vitro Drug Release Test and Morphology Evaluation

The cumulative release curves of the HA/PBS/MPT composites and MPT-10% samples are presented in Figure 5, clarifying that the introduction of HA could increase the release rate of MPT in the PBS matrix (Table 4). In the initial 50 h, the MPT release amount of MPT-10% was about 32%, and that of MPT-HA-2%, MPT-HA-5%, and MPT-HA-10% was 49.8%, 58.4%, and 72.5%, respectively. A similar tendency was also detected in 150 h and 300 h. According to the above hydrophilic contact angle test and crystal evaluation, the introduction of HA could increase the hydrophilicity of HA/PBS/MPT composites and decrease the crystal integrity of PBS, which was conductive to the PBS solution infiltrating into the interior of the PBS matrix to dissolve the MPT. That could be the reason that the MPT release rate increased with an increase in the HA content.

The morphology of the HA/PBS/MPT composites during the immersion in the PBS solution was also observed using SEM (Figure 6). After immersion for different times, porous structures were formed on the section of the composite samples due to the dissolution of MPT, and the pore size increased gradually with the extension of the immersion time. As Appendix A present, the pore diameter of MPT-HA-10% was about 5.1 μm, 8.3 μm, and 10.2 μm after immersion for 2 h, 12 h, and 36 h, respectively. Furthermore, the HA content played a significant role in pore formation. Under the same immersion time, the pore size of the samples with a high HA content was significantly larger than that of the samples with a relatively low HA content. After immersion of 2 h, the pore diameters of MPT-HA-2%, MPT-HA-5%, and MPT-HA-10% were about 2.5 μm, 3.8 μm, and 5.1 μm, respectively. And immersion for 36 h, they were about 4.1 μm, 4.9 μm, and 10.2 μm, respectively (Appendix A). From Figure 6, it can be seen that, with an increase in pore size, the HA aggregations were exposed, which was clarified by an analysis of the SEM associated with EDS (Appendix A).

Associated with the in vitro release test and morphology evaluation, we could obtain that, for the HA/PBS/MPT composites, the introduction of HA increased the hydrophilicity of the HA/PBS/MPT composites and the MPT release rate, forming pores with a larger size. Thus, the HA content could regulate the MPT release rate and pore size of the HA/PBS/MPT composites at the same time. The more HA content, the faster the MPT release and the larger the pore size.

### 3.5. In Vitro Cytocompatibility Evaluation

TRITC Phalloidin/DAPI staining was carried out for the in vitro cytocompatibility evaluation of the samples co-cultured with MC3T3-E1 cells after 24 h, and the fluorescence images of the MC3T3-E1 cells are presented in Figure 7. DAPI was the fluorescent dye used for labeling the nucleus, and TRITC labeled phalloidin could label the cytoskeleton. The blue bright spot and red area in the image indicate the nucleus and cytoskeleton of the MC3T3-E1 cells, respectively. The MC3T3-E1 cells basically maintained their original spindle shape after co-culturing on the surface of PBS for 24 h, and did not show obvious plasmodesmata (Figure 7a). For the PBS/HA composites (Figure 7b–d), it could be clearly observed that the proliferation of the MC3T3-E1 cells was better than that of PBS. With an increase in the HA content, more MC3T3-E1 cells could be seen under the same multiple, indicating that the introduction of HA was conducive to the growth and proliferation of the MC3T3-E1 cells.

For the HA/PBS/MPT composites, it could be obviously observed that the cell proliferation rate was significantly higher than that of the PBS/HA composites with the same HA addition (Figure 7e–h). The MC3T3-E1 cells co-cultured on the surface of the HA/PBS/MPT composites showed an obvious regional accumulation and a large amount of plasmodesmata. Similar to the PBS/HA composites, the cell proliferation of the HA/PBS/MPT composites gradually improved with an increase in HA content. The MC3T3-E1 cells co-cultured with MPT-HA-10% could even proliferate to cover the whole observation field after 24 h of culturing (Figure 7h).

For the HA/PBS/MPT composites, porous structures could be clearly observed (white dotted circle) on the surface due to the dissolution of MPT, consistent with the above morphology evaluation in the PBS solution. Large numbers of MC3T3-E1 cells were distributed inside and around the pores, indicating that the MC3T3-E1 cells could better adhere, grow, and proliferate near the pores. This could be the reason that the HA aggregations exposed by the dissolution of MPT could significantly promote the growth and proliferation of the MC3T3-E1 cells.

## 4. Conclusions

In this work, we fabricated HA/PBS/MPT composites using a simple and effective melt-blending method, aiming to provide an effective and convenient strategy for achieving the optimization of the porous structure formation and drug release regulation of bone scaffolds. The introduction of HA significantly enhanced the hydrophilicity of the PBS matrix. Specifically, when the HA addition amount was 10%, the water contact angle of MPT-HA-10% was 24.9°, which was approximately 36° lower than that of the MPT-10% system. Additionally, it disrupted the crystalline integrity of PBS. Both effects can enhance the permeability of the matrix, facilitating the penetration of PBS into the composite matrix and accelerating the release of MPT. Accompanied with the release of MPT, porous structures were formed in situ and the HA inside the matrix was exposed. The higher the content of HA, the faster the MPT release rate and the larger the pore size. With an increase in the content of HA in the ternary system, the drug release rate of the loaded samples gradually accelerated. When released for 50 h, the cumulative drug release of MPT-HA-10% was approximately 36% higher than that of the MPT-10% system without added HA. When released for 300 h, the difference in the cumulative drug release between MPT-HA-10% and MPT-10% stabilized at 23%. The in vitro cytocompatibility evaluation indicated that the porous structure of the HA/PBS/MPT composites promoted the adherence, growth, and proliferation of the MC3T3-E1 cells, due to the HA exposed around the pores. Thus, the porous structure, MPT release rate, and cell induction ability of the HA/PBS/MPT scaffold were flexibly and effectively regulated by the composition at the same time. This work might provide a convenient and robust strategy for the fabrication of bone scaffolds with controllable drug release and porous structures. Additionally, it establishes high-performance bone scaffolds capable of achieving efficient drug release and synergistic bone regeneration induction. These findings hold potential value in the field of clinical medicine for repairing bone injuries.

## Figures and Tables

**Figure 1 polymers-15-04205-f001:**
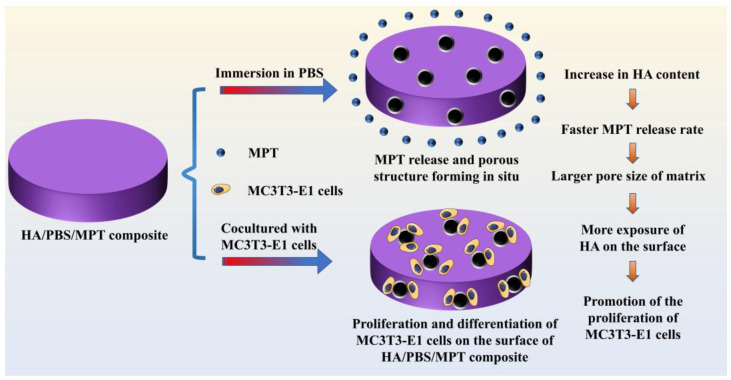
Schematic diagram illustrating hydroxyapatite/poly (butylene succinate)/metoprolol tartrate composites with controllable release behavior and a porous structure for bone scaffold application.

**Figure 2 polymers-15-04205-f002:**
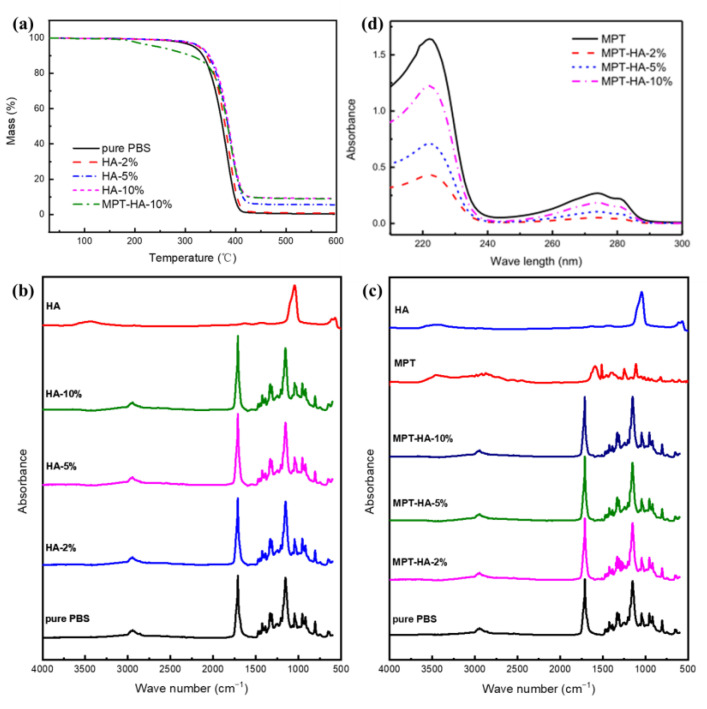
TGA curves (**a**), FT-IR (**b**,**c**) and UV-vis spectra (**d**) of HA, PBS, MPT, PBS/HA composites and HA/PBS/MPT composites.

**Figure 3 polymers-15-04205-f003:**
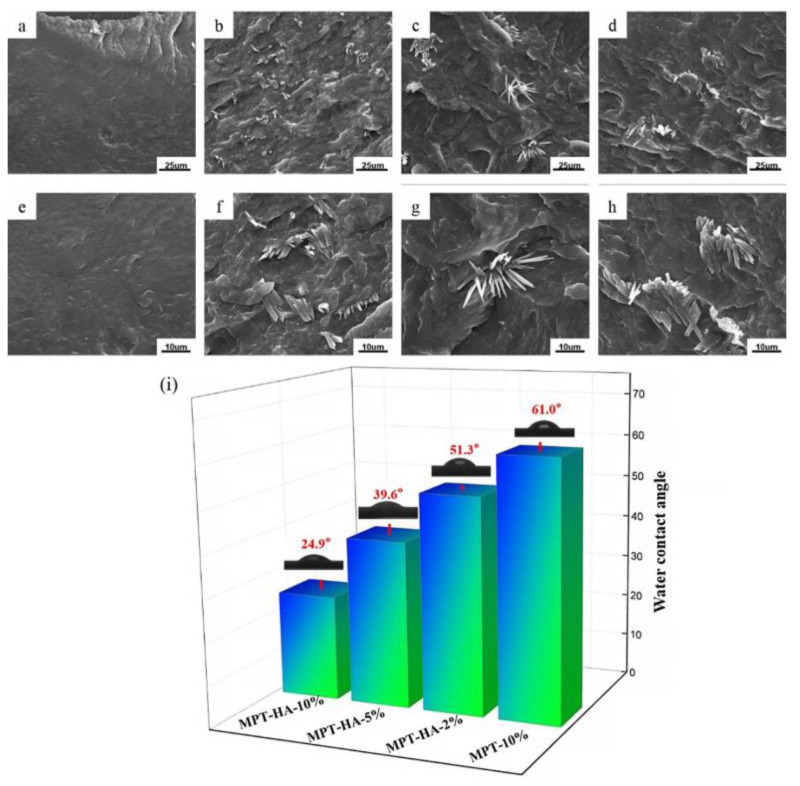
The morphology of PBS and PBS/HA composites ((**a**,**e**) PBS, (**b**,**f**) HA-2%, (**c**,**g**) HA-5%, and (**d**,**h**) HA-10%); water contact angles of MPT-10% and HA/PBS/MPT composites (**i**).

**Figure 4 polymers-15-04205-f004:**
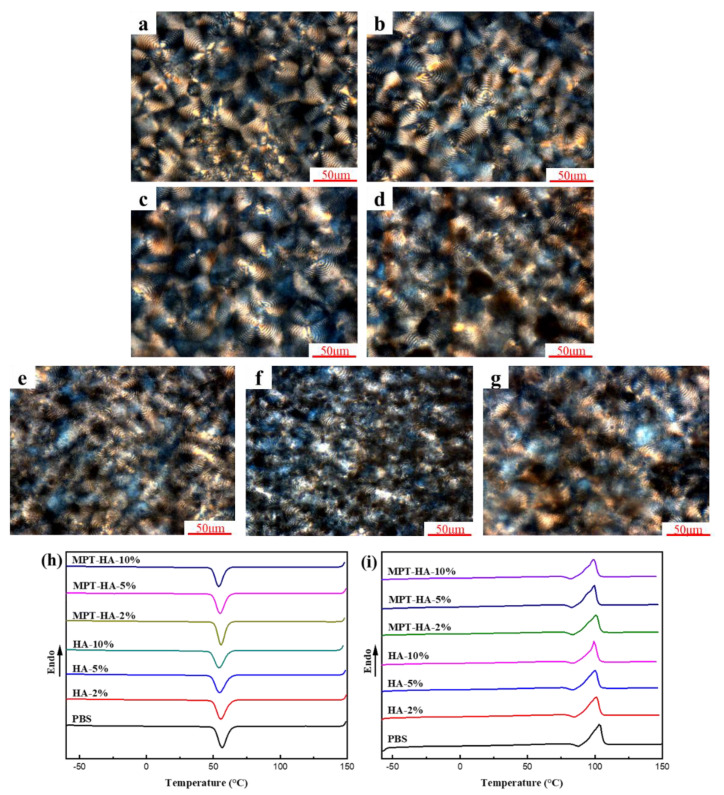
POM photographs of PBS, PBS/HA and HA/PBS/MPT composites ((**a**) PBS, (**b**–**d**) PBS/HA (**b**) HA-2%, (**c**) HA-5%, and (**d**) HA-10%), and (**e**–**g**) PBS/MPT/HA EUR MPT-HA-2%, (**f**) MPT-HA-5%, (**g**) MPT-HA-10%), scale bar: 50 μm)), and DSC crystallizing (**h**) and melting (**i**) curves of PBS, PBS/HA, and PBS/MPT/HA composites.

**Figure 5 polymers-15-04205-f005:**
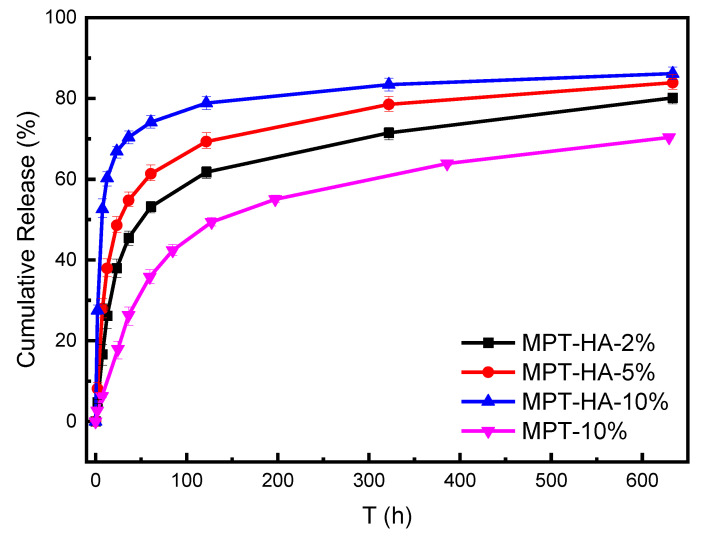
Cumulative release of MPT from HA/PBS/MPT composites and MPT-10% samples at 37 °C in PBS solution.

**Figure 6 polymers-15-04205-f006:**
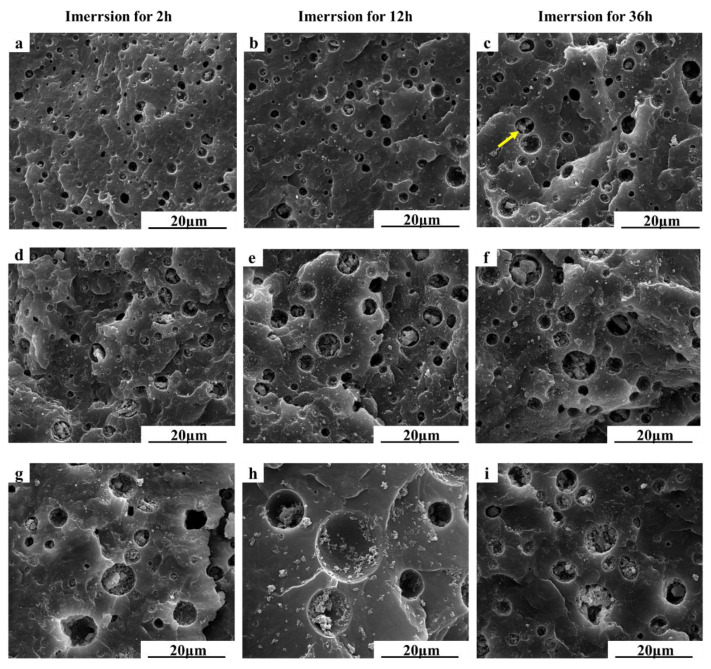
Evolution of fractured morphology of HA/PBS/MPT composites during immersion in PBS for different times ((**a**–**c**) MPT-HA-2%, (**d**–**f**) MPT-HA-5%, and (**g**–**i**) MPT-HA-10%) (yellow arrow indicating the HA aggregations).

**Figure 7 polymers-15-04205-f007:**
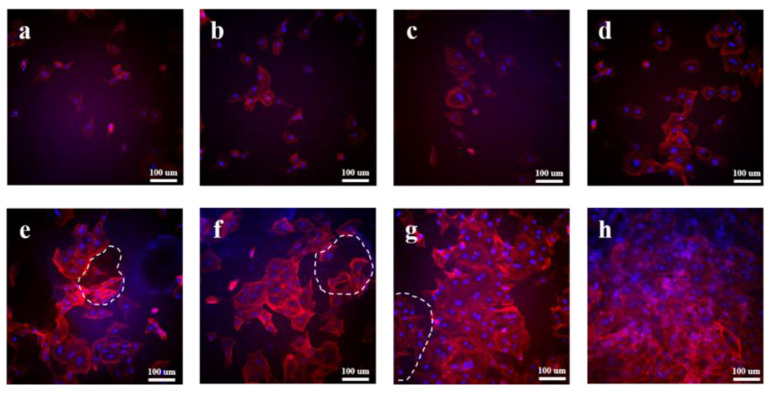
Fluorescent microscopy analysis of TRITC Phalloidin/DAPI stained MC3T3-E1 cells seeded on (**a**) PBS, (**b**) HA-2%, (**c**) HA-5%, (**d**) HA-10%, (**e**) MPT-HA-2%, (**f**) MPT-HA-5%, and (**g**,**h**) MPT-HA-10%, respectively.

**Table 1 polymers-15-04205-t001:** The descriptors and composition of composite samples.

Specimen	PBS, wt%	HA, wt%	MPT, wt%
PBS	100	/	/
HA-2%	98	2	/
HA-5%	95	5	/
HA-10%	90	10	/
MPT-HA-2%	88	2	10
MPT-HA-5%	85	5	10
MPT-HA-10%	80	10	10
MPT-10%	90	/	10

The “/” symbol indicates that the substance is not present in the sample.

**Table 2 polymers-15-04205-t002:** The initial decomposition temperature (Td) and residual quantity (R) for PBS, PBS/HA composites, and MPT-HA-10%.

Specimen	Td, °C	R, %
PBS	348.4	0.3
HA-2%	355.1	1.1
HA-5%	358.1	5.5
HA-10%	359.9	9.1
MPT-HA-10%	181.2	8.8

**Table 3 polymers-15-04205-t003:** The crystallization temperature (Tc), melting temperature (Tm), heat of fusion (ΔHm), and the degree of crystallinity (Xc) for PBS, PBS/HA, and PBS/MPT/HA composites.

Specimen	Tc, °C	Tm, °C	ΔHm, J/g	Xc, %
PBS	56.9	102.1	55.0	27.5
HA-2%	56.4	101.1	52.3	26.7
HA-5%	55.2	100.2	49.4	26.0
HA-10%	55.0	99.6	45.2	25.1
MPT-HA-2%	56.5	100.8	46.3	26.3
MPT-HA-5%	55.1	100.4	43.7	25.7
MPT-HA-10%	54.8	99.3	39.4	24.6

**Table 4 polymers-15-04205-t004:** Cumulative release amount of t HA/PBS/MPT composites under different time segments.

Specimen	5 h	20 h	50 h	150 h	300 h
MPT-10%	4.7%	15.1%	32.0%	51.4%	59.9%
MPT-HA-2%	10.9%	34.5%	49.8%	63.1%	70.4%
MPT-HA-5%	18.3%	45.6%	58.4%	70.6%	77.5%
MPT-HA-10%	40.6%	65.3%	72.5%	79.5%	82.9%

## Data Availability

The data presented in this study are available on request from the corresponding author.

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
