# Peer review of "Hydroxyapatite/Poly (Butylene Succinate)/Metoprolol Tartrate Composites with Controllable Drug Release and a Porous Structure for Bone Scaffold Application"

_polymers, 2023, doi:10.3390/polym15214205_

Round 1

Reviewer 1 Report

The paper is describing a composite that has in situ pore forming ability which creates pores that can allow better cell induction ability. HA facilitated MPT release with exposure of HA around the pores resulting in the better cell induction ability. Pores aided in penetration of phosphate buffer solution into composite. Performance was altered according to composition

The paper was reasonably clearly discussed but needs improvement especially when it would be read by a wide audience. 

The English is moderately OK but has strange non grammatical constructions which lessens the impact of the paper. In the abstract there are English errors like "adhere, grow and ...." which should be nouns like "adhesion" and "growth"

The word "tough" seems inappropriate in a formal scientific paper and it seems misused in places

In the introduction I dont know what "With the in depth of clinical application" is meant to mean? 

Figure 1 caption ....."illustrating of " is the wrong thing to sayh

Figure 1 also needs some clarification in the drawing ...what do the ascending arrows mean? The diagram is too cryptic to understand...they need to clarify or redraw it

"The abbreviation and composition ...." ....."abbreviation of what?? The statement makes no sense ...do you mean the "descriptors"??

Table 1  ..the "/" symbol is not explained ...what does it mean exactly? Do we have to guess? ...scientific papers should not involve guessing but should be precise ..I am presuming it means "not applicable"? 

In the DSC ...what is the reason for "eliminating thermal history"? It is not clear. 

For contact angle measurements it was not completely clear why samples needed to be quenched in liquid nitrogen ...remember please that the readership of such articles will necessarily be across a broad disciplinary base so some things need to be better explained for their benefit

Figure 2 ...FTIR spectra are extremely small!!  Not even a magnifying glass can show these?  The figures need to be much larger . In particular HA's spectrum is just a small barely observable peak. Please enlarge ALL diagrams to facilitate observation

The term "dispersion absorption peak" does not make sense ...do you mean the lattice OH stretching peak in HA?

Morphology and hydrophilicity ....comments ....HA aggregation is spoken o but it is discussed with very little depth other than just to observe it ...why is this important? Please explain the significance ...

The word "serious" is used ....does this mean the composites with the aggregation are of limited value ..?  Word choice is important

The sentence beginning ..."as far as we knew ...." seems an inappropriate comment to make in a scientific paper ...it sounds speculative and like "gossip"  ...what is the evidence and the citing references??? 

Crystal morphology section ...

This needs better explanation for the wider audience reading it ... the term "steric effect of HA" is stated but with no explanation. Please explain the significance?  

Use of words like "badly damaged" sounds like the composites made are of very limited value and are no good as specimens ...can better descriptors be used to explain the significance of the decrease of the crystal  integrity of PBS?  This whole section needs better describing and removal of awkward English which leads to misunderstanding of the material described.  What is the scientific significance and what information does this give us about how the composites will behave? 

Figure 4 ...plots too small ...

What factors (Table 4) cause change in deltaHm??

What is a hydrophilicity analysis???  Is this a term made up by the authors or is it a real scientific term? 

.."with the increase of pore sizxe, the HA aggregations were exposed"  ...the word "aggregations" is not explained again....See my earlier comment 

The phrase "an effect factor" ...is incorrect English and needs revising 

"after co-cultured" ...incorrect English 

Conclusions ...

beneficial to the "adhere, grow" ...etc ...incorrect English 

Comments on English already described ...

Please have a native English language speaker check on English construction

Reviewer 2 Report

Dear authors

Thank you so much for your contribution and submission to polymers. I would like to suggest you to improve the image quality and enlarge as its very difficult to observe clearly specially Fig. 4 (a,b,c,d,e,f an g), 5, 6

I believe the language is considerable but still i suggest to send to a scientific language editor to furnish it as journal standard. 

Reviewer 3 Report

1. The author should include the novelty of the work in the abstract with qualitative and quantitative results.

2. In the last paragraph of the introduction, the Author needs to clearly state the novelty of this paper together with detailed aims and future prospects of this study. In particular, there are several recent reviews in the literature on this field. The author needs to explain these together in this review.

3. What is the hypothesis behind this work and the rationale behind using those two combinations? What is the reason for choosing the various wt.% of HA and MPT? Why MPT is chosen as the model drug?

4. Authors are expected to explain in detail the in situ pore form technique used in the study so that the readers can replicate them. 

5. How did you ensure uniform thickness of the scaffold?

6. What is the significance of the TGA studies in this work? How is it related to biomedical applications especially in bone tissue engineering?

7. How about the mechanical strength of the scaffold? Why it is not reported?

8. The results are presented well. But it would be important and useful to explain with rigour the scientific reasons/explanations behind those observations made.

9. The conclusions could be revised by including the key observations made together with short explanations. It would also help if a general outlook is given highlight the broader conclusion(s).

Round 2

Reviewer 1 Report

Amendments and revisions are satisfactory